# Investigation of the Role of Healthy and Sick Equids in the COVID-19 Pandemic through Serological and Molecular Testing

**DOI:** 10.3390/ani12050614

**Published:** 2022-02-28

**Authors:** Kaila O. Y. Lawton, Rick M. Arthur, Benjamin C. Moeller, Samantha Barnum, Nicola Pusterla

**Affiliations:** 1Department of Medicine and Epidemiology, School of Veterinary Medicine, University of California, Davis, CA 95616, USA; kolawton@ucdavis.edu (K.O.Y.L.); smmapes@ucdavis.edu (S.B.); 2School of Veterinary Medicine, University of California, Davis, CA 95616, USA; rmarthur@ucdavis.edu; 3KL Maddy Equine Analytical Chemistry Laboratory, School of Veterinary Medicine, University of California, Davis, CA 95616, USA; bcmoeller@ucdavis.edu; 4Department of Molecular Biosciences, School of Veterinary Medicine, University of California, Davis, CA 95616, USA

**Keywords:** SARS-CoV-2, horses, nasal secretions, blood, qPCR, ELISA, sick equids, healthy horses

## Abstract

**Simple Summary:**

The objective of the present study was to determine if horses are susceptible to SARS-CoV-2. Nasal swabs from 667 equids with acute onset of fever and respiratory signs were tested by qPCR for SARS-CoV-2. Further, 633 serum samples collected from a cohort of 587 healthy racing Thoroughbreds with possible exposure to humans with SARS-CoV-2 infection were tested for antibodies to SARS-CoV-2 using an ELISA targeting the receptor-binding domain of the spike protein. All 667 horses with fever and respiratory signs tested qPCR-negative for SARS-CoV-2. A total of 35/587 (5.9%) Thoroughbred racing horses had detectable IgG antibodies to SARS-CoV-2. While horses appear to be susceptible to SARS-CoV-2 when in close contact with humans with SARS-CoV-2 infection, clinical disease was not observed in the study horses. Experimental challenge studies using pure inocula are needed in order to study the clinical, hematological, molecular, and serological features of adult horses infected with SARS-CoV-2.

**Abstract:**

More and more studies are reporting on the natural transmission of SARS-CoV-2 between humans with COVID-19 and their companion animals (dogs and cats). While horses are apparently susceptible to SARS-CoV-2 infection based on the homology between the human and the equine ACE-2 receptor, no clinical or subclinical infection has yet been reported in the equine species. To investigate the possible clinical role of SARS-CoV-2 in equids, nasal secretions from 667 horses with acute onset of fever and respiratory signs were tested for the presence of SARS-CoV-2 by qPCR. The samples were collected from January to December of 2020 and submitted to a commercial molecular diagnostic laboratory for the detection of common respiratory pathogens (equine influenza virus, equine herpesvirus-1/-4, equine rhinitis A and B virus, *Streptococcus equi* subspecies *equi*). An additional 633 serum samples were tested for antibodies to SARS-CoV-2 using an ELISA targeting the receptor-binding domain of the spike protein. The serum samples were collected from a cohort of 587 healthy racing Thoroughbreds in California after track personnel tested qPCR-positive for SARS-CoV-2. While 241/667 (36%) equids with fever and respiratory signs tested qPCR-positive for at least one of the common respiratory pathogens, not a single horse tested qPCR-positive for SARS-CoV-2. Amongst the racing Thoroughbreds, 35/587 (5.9%) horses had detectable antibodies to SARS-CoV-2. Similar to dogs and cats, horses do not seem to develop clinical SARS-CoV-2 infection. However, horses can act as incidental hosts and experience silent infection following spillover from humans with COVID-19. SARS-CoV-2-infected humans should avoid close contact with equids during the time of their illness.

## 1. Introduction

Epidemiological work in the field of SARS-CoV-2 has focused on the human–animal interface in order to identify animal species, which could act as reservoirs and intermediate hosts [1]. Understanding the host range for SARS-CoV-2 is important in order to control the ongoing pandemic and to protect populations of wild and domestic animals in their native habitat and under human care, respectively. The best-documented evidence for susceptibility of any animal species comes from detecting SARS-CoV-2 under natural conditions or proof of active viral transmission between infected and susceptible in contact animals. While experimental inoculations of selected animal species are needed to document viral kinetics and risk of viral transmission, such protocols only mirror, but never reproduce, natural conditions. The predictive susceptibility of animals has also been based on computational modelling of their angiotensin-I-converting enzyme 2 (ACE-2), a key receptor for SARS-CoV-2 [2]. ACE-2 serves as a functional receptor for the spike protein of SARS-CoV-2 [3]. Cross-species infections can occur when a coronavirus adapts to a new host in part through the mutation of the spike protein, shown to enhance the binding affinity for ACE-2 [4]. Using comparative genomic approaches and protein structural analysis, Damas and colleagues [2] determined the conservation of ACE-2 and its potential to be used as a receptor by SARS-CoV-2 in 410 vertebrate species. Their results showed that mammals fell into low to high binding categories, with *equus caballus* and *equus asinus* displaying a low binding score category for SARS-CoV-2. 

The close interactions of domestic animals with humans worldwide make determining their susceptibility an urgent need. Human-to-animal transmissions of SARS-CoV-2 have been documented in dogs, cats, tigers, lions and minks [5,6,7]. The role of equids in the COVID-19 pandemic has remained poorly investigated. Horses are potentially susceptible to SARS-CoV-2 based on the binding affinity and stability between ACE-2 and the receptor-binding domain of the S protein [8,9]. Considering the large number of equids globally and the direct or indirect contact these animals have with humans, information pertaining to their susceptibility to SARS-CoV-2 and their role in virus transmission is needed. Therefore, the aims of the present study were to determine if SARS-CoV-2 could be detected in nasal secretions of equids with acute onset of fever and respiratory signs using qPCR and to investigate the seroprevalence against SARS-CoV-2 in a cohort of racing horses with possible exposure to humans with SARS-CoV-2 infection. 

## 2. Materials and Methods

### 2.1. Study Population and Sampling

Nasal fluid samples from 667 equids with acute onset of upper airway infection were enrolled in the study. The same samples were used to investigate three newly identified equine parvoviruses in a recent study [10]. The respiratory secretions were submitted to a commercial diagnostic laboratory from 1 January 2020 to 31 December 2020 for the molecular detection of common respiratory pathogens, including equine influenza virus (EIV), equine herpesvirus-1/-4 (EHV-1/-4), equine rhinitis A and B virus (ERVs) and *Streptococcus equi* subspecies *equi* (*S. equi*).

Six hundred and thirty-three serum samples from 587 racing Thoroughbred horses from California, collected from 10 July 2020 to 12 September 2020, were available for antibody testing against SARS-CoV-2. The blood samples had been collected as part of the routine medication testing program established by the California Horse Racing Board. For the majority of the racing horses, only one serum sample was available, while 2 and 3 consecutive serum samples were available for 36 and 5 horses, respectively. The samples were stored at −80 °C until testing. The period of sample collection coincided with a known outbreak of COVID-19 at the sampling location with 22 asymptomatic track personnel testing qPCR-positive for SARS-CoV-2 (https://www.washingtonpost.com/sports/2020/07/16/del-mar-cancels-racing-after-22-positive-covid-19-tests-among-jockeys-track-workers, accessed on 1 November 2021). Because of confidentiality issues, only the age and sex of the 587 racing Thoroughbred horses were made available to the researchers performing the testing by sample identification numbers. 

Serum samples collected from 88 healthy adult horses in 2015 (pre-COVID-19 pandemic) and stored at −80 °C until testing served as negative control to establish the cutoff value for the ELISA. Serum samples from 24 horses with previously confirmed ECoV infection were available to test possible cross-reactivity using the SARS-CoV-2 ELISA [11]. 

### 2.2. Quantitative PCR Analyses

Nasal fluid samples from 667 horses with acute onset of fever and respiratory signs were tested for the presence of EIV, EHV-1/-4, ERVs and *S. equi* as previously reported [10,12,13]. Primers and probes targeting the S gene of SARS-CoV-2 were designed following BLAST analysis of published sequences from GenBank (www.ncbi.nlm.nih.gov/genbank, accessed on 1 March 2020) (Table 1). Amplification of the target gene was performed using a commercial thermocycler/fluorometer (QuantStudio 5, Applied Biosystems, Foster City, CA, USA). The standard amplification conditions were as follows: 2 min at 50 °C, 10 min at 95 °C, and 40 cycles of 15 s at 95 °C and 60 s at 60 °C. Each PCR reaction for the 6 equine respiratory pathogens and SARS-CoV-2 contained a commercially available mastermix (Universal TaqMan Mastermix with AmpErase UNG, Applied Biosystems, Foster City, CA, USA), 0.625 U of AmpliTaq Gold, 400 nM of each primer and 80 nM of the respective TaqMan probe, and 1 μL of DNA or 5 μL of cDNA sample for a total volume of 12 μL. For the SARS-CoV-2 qPCR assay, a standard curve was generated using plasmid containing the target sequence (Table 1). The amplification efficiency of the SARS-CoV-2 qPCR assay was calculated from the slope using the formula E = 10^(−1/slope). The amplification efficiency was 99% for the spike protein gene of SARS-CoV-2, indicating a very high analytical sensitivity. The detection limit for the SARS-CoV-2 qPCR assay was 13 genome equivalents when the cDNA was purified from nasal secretions. The quality and efficiency of nucleic acid extraction were determined by targeting an equine housekeeping gene as previously described [12].

### 2.3. Serology

Antibody detection was performed by adapting an assay initially described by Zhao and colleagues [14]. The assay targets the S protein, specifically the immunodominant receptor-binding domain (RBD). Microtiter plates were coated with 100 µL of recombinant SARS-CoV-2 RBD of the spike protein (ThermoFisher Scientific, Waltham, MA, USA) diluted in coating buffer (Bethyl Laboratories Inc., Montgomery, TX, USA) at a concentration of 100 ng/mL. Plates were then covered and stored at 4 °C overnight. Serum samples from the study horses previously stored at −80 °C were thawed overnight at 4 °C. On the day of the analysis, the coated plates were washed 4 times with 200 µL of wash buffer (Bethyl Laboratories Inc., Montgomery, TX, USA) per well and gently tapped until dry. Then, each well received 90 µL of sample dilution buffer (Bethyl Laboratories Inc., Montgomery, TX, USA) and 10 µL of serum; each sample was run in singlet. Optimal S protein and serum dilutions were determined prior to assay validation using standard checkboard titration procedures. After the serum samples were loaded into the wells, the plates were covered and wrapped in aluminum foil and incubated for 2 h at room temperature on a titer plate shaker. Thereafter, the plates were washed 4 times, and 100 µL of diluted anti-horse IgG horseradish peroxidase conjugate (dilution of 1:120,000 in 2% milk; Sigma Aldrich, St. Louis, MO, USA) was added. This step was followed by 1 h incubation as mentioned above. After washing the plate 4 times, 100 µL of enzyme substrate (Bethyl Laboratories Inc., Montgomery, TX, USA) was added to each well. The plate was then incubated at room temperature for 10 min. As a final step, 50 µL of stop solution (4.89 mL of 98% sulfuric acid diluted with 495 mL of distilled water) was added to each well. The optical density (OD) was measured at 450 nm in a microplate photometer (Spectramax 250, Molecular Devices Corp., Sunnyvale, CA, USA). The OD was measured within 15 min of adding the stop solution. Cut-off values were determined as six times the standard deviations above the mean value of reactivity of 88 seronegative samples from a pre-COVID-19 cohort of healthy adult horses [15]. Because of the inability to test the serum samples using the reference standard of virus neutralization, seropositive serum samples determined via the ELISA targeting the SARS-CoV-2 RBD of the spike protein were defined as suspect positive.

### 2.4. Statistical Analyses

Demographic and clinical information from horses with upper airway infection, healthy racing horses and healthy controls was evaluated using descriptive analyses. All statistical analyses were performed using Stata Statistical Software (College Station, TX, USA), and statistical significance was set at *p* < 0.05.

## 3. Results

Demographic and clinical information from horses with acute onset of fever and respiratory signs was previously reported [10]. Briefly, the population ranged in age from 1 month to 34 years (median 9 years), with greater numbers of males (61%) compared to females (39%). A variety of breeds were represented and included Quarter Horse (37%), Warmblood (14%), Thoroughbred (10%), pony breed (6%), Arabian (5%), Paint Horse (4%) and other breeds (22%). The three most commonly reported clinical signs included fever (97%, range 38.6 to 41.4 °C, median 39.4 °C), nasal discharge (74%) and coughing (46%). Common respiratory pathogens were detected in 241/667 (36%) sick equids (81 EIV, 61 *S. equi*, 50 EHV-4, 36 ERVs, 13 EHV-1). Overall, not a single equid tested qPCR-positive for SARS-CoV-2 by qPCR. 

The 88 pre-COVID-19 control horses were composed of 53 males (60%) and 35 females (40%) ages 2 to 12 years (median 4.6 years). The OD for the 88 pre-COVID-19 horses ranged from 0.030 to 0.358 (median 0.122, Figure 1). The cutoff value for a suspect positive SARS-CoV-2 ELISA was set at an OD value of ≥0.507. The population of ECoV-seropositive horses was composed of 13 males (54%) and 11 females (46%) aged 4–22 years (median 17.5 years). All 24 ECoV-seropositive horses were seronegative for SARS-CoV-2 with OD values ranging from 0.081 to 0.384 (median 0.137). The population of 587 racing horses was composed of 335 males (57%) and 252 females (43%) aged 2–7 years (median 3 years). The OD for the 633 serum samples ranged from 0.004 to 1.298 (median 0.091). A total of 40/633 serum samples (6.3%) were considered suspect seropositive for SARS-CoV-2 by ELISA with an OD ≥ 0.507 (range 0.510 to 1.298, median 0.911; Figure 1). The 40 SARS-CoV-2 suspect seropositive serum samples originated from 35/587 horses (5.9%). Thirty-one horses had a single SARS-CoV-2 suspect seropositive sample, three horses had two suspect seropositive samples (days between serum collections ranged from 28 to 44 days) and one horse had three suspect seropositive samples (days from first to third serum collection was 46 days). Amongst the thirty-one horses with a single SARS-CoV-2 suspect seropositive sample, four horses showed seroconversion between two sample collection time points (days between serum collections ranged from 22 to 41 days). 

## 4. Discussion

It has been shown that various domestic animal species, including cats, dogs and farmed minks, are susceptible to SARS-CoV-2 infection under natural and experimental conditions [16]. While most of these animal species are permissive to infection, clinical pathology does not always mimic disease observed in humans. Many factors, including genetic diversity, age, comorbidity, expression of ACE-2 receptor and pre-existing diseases, have been shown to modulate disease form [17,18]. Little is known about the prevalence of SARS-CoV-2 in large domestic animal species such as equids. In a serological survey of SARS-CoV-2 in different species of animals from China, no antibodies specific to SARS-CoV-2 were found in serum samples from 18 horses [19]. Therefore, the aim of this study was to investigate the susceptibility to SARS-CoV-2 in equids with acute respiratory disease and in healthy racehorses in close contact with humans with asymptomatic SARS-CoV-2. 

The lack of detectable SARS-CoV-2 by qPCR in nasal secretions of 667 horses with acute onset of fever and respiratory signs is in agreement with an investigation performed by IDEXX Reference Laboratories on over 6000 canine, feline and equine specimens tested for SARS-CoV-2 by qPCR from mid-February to mid-April, 2020 (https://www.idexx.com/en/veterinary/reference-laboratories/overview-idexx-sars-cov-2-covid-19-realpcr-test, accessed on 1 November 2021). A recent study evaluating nasal and nasopharyngeal swabs and feces from 34 healthy Italian Trotters with recent contact with SARS-CoV-2 breeders showed no detection of SARS-CoV-2 by qPCR [20]. Another study evaluating the susceptibility of common domestic livestock showed no clinical disease, no nasal and fecal viral shedding determined by qPCR and no virus isolation from respiratory tissues in a single horse following intranasal administration of 6.3 log_10_ plaque-forming units SARS-CoV-2 virus strain 2019-nCoV/USA-WA1/2020 [21]. The reason for negative SARS-CoV-2 qPCR results in the present study population may relate to the lack of disease expression in equids, similar to other domestic animals [6]. Various studies have demonstrated that SARS-CoV-2 infection in companion animals (dogs and cats) is mostly detected in animals living in households with at least one SARS-CoV-2-infected human. The reported frequencies of SARS-CoV-2 infection in dogs and cats confirmed by molecular methods ranges from 0–28% and 0–40%, respectively [22,23,24,25,26,27]. Close contact of dogs and cats with their SARS-CoV-2-infected owners, especially sharing the bed with an infected human, was recently determined as the main risk factor for transmission [25]. Contact between equids and owners, trainers and barn workers is generally limited in time, with greater physical distances kept between handlers and horses, and contact often occurs in the outdoors. The latter management and husbandry practices are less likely to promote SARS-CoV-2 transmission between SARS-CoV-2-infected humans and equids. To study the impact of SARS-CoV-2-infected horse owners on their horses, prospective longitudinal studies are needed in order to sample horses at regular intervals once horse owners have been diagnosed with COVID-19. 

Studies focusing on animals with possible exposure to people with COVID-19 have the potential to quantify the risk of transmission between humans shedding SARS-CoV-2 and susceptible animals. The known asymptomatic qPCR-positive test results of track personnel for SARS-CoV-2 at the racing location represented a unique opportunity to determine potential spillover from infected humans to race horses. The 633 convenience blood samples were collected over a 9-week period, covering a period when racing was cancelled due to the human positive cases. The study results showed that 5.9% of tested horses had antibodies against the RBD of SARS-CoV-2. Due to the small volume of serum available for each racehorse, the samples were run in singlets and the results could not be confirmed via retesting. Further, another limitation was the inability to confirm ELISA positive results using the reference standard of virus neutralization. These limitations may have impacted true seroprevalence against SARS-CoV-2. This relatively high percentage of suspect seropositivity in horses could be related to the large number of infected jockeys and track workers having contact with the racing horses. Of interest was the observation that 4 Thoroughbred racing horses seroconverted to SARS-CoV-2 during the study period. However, the study design does not allow for the determination of whether human-to-horse or horse-to-horse transmission occurred. Nevertheless, to the authors’ knowledge, this is the first report showing the exposure of horses with SARS-CoV-2 secondary to spillover from asymptomatic humans. Laboratory-based qPCR is the recommended test for diagnoses of acute cases, while serological tests are important to define epidemiological questions, such as exposure rate [28]. The serological platform used for this study was based on the detection of the RBD of SARS-CoV-2, shown to be one of the most specific antigens [14,29]. Further, the RBD-specific SARS-CoV-2 did not show any cross-reactivity with the closely-related ECoV, ruling out any false-positive results. Studies assessing seroprevalence in companion animals living in households with SARS-CoV-2-infected owners reported seropositivity rates of 3.4–23.5% for dogs and 4–43.8% for cats [23,24,25,27,30,31]. Because the SARS-CoV-2 shedding status of jockeys and track workers attending every single study horse was unknown, it was impossible to determine the time of infection. Experimental studies using susceptible animals such as cats and documented cat-to-cat transmissions have shown seroconversion occurring as early as 11–12 days post-infection [32]. A similar time to seroconversion can be assumed for other susceptible animal species such as equids. Limitations of the study relate to the lack of longitudinal data from the same horses during the study period, as well as the inability to test nasal or nasopharyngeal secretions for SARS-CoV-2 by qPCR. Further, without sequence information of the SARS-CoV-2 involved in horse and human infections, the authors cannot conclude that horses were infected with the same virus responsible for asymptomatic COVID-19 in humans.

During the monitoring period, no outbreak of a respiratory disease was reported in the racing horses, suggesting that horses with antibodies to SARS-CoV-2 likely experienced subclinical infection. Horses do apparently remain subclinical following infection with SARS-CoV-2. The susceptibility to developing COVID-19 in companion animals is a complex interplay between various viral and host factors [33]. While data is limited on the susceptibility to SARS-CoV-2 infection in domestic animals, it appears that equids are incidental hosts because of occasional SARS-CoV-2 spillover from humans. However, continuous surveillance is necessary in order to monitor the possible transmission of SARS-CoV-2 infection in equids. From a biosecurity perspective, it is highly recommended that humans with clinical and asymptomatic SARS-CoV-2 infection avoid close contact with any companion animals. 

## 5. Conclusions

In conclusion, our results show that equids are susceptible to natural SARS-CoV-2 infections. While SARS-CoV-2 could not be detected via qPCR in nasal secretions of horses with acute onset of fever and respiratory signs, antibodies specific to SARS-CoV-2 were found in 5.9% of healthy racing Thoroughbreds in close contact with humans with asymptomatic SARS-CoV-2 infection. Similar to other companion animals, horses appear to be incidental hosts because of occasional SARS-CoV-2 spillover from humans. From an epidemiological standpoint, it is important to continue to monitor the possible transmission of SARS-CoV-2 infection in equids and other domestic animals and to emphasize the risk of SARS-CoV-2 transmission from humans with clinical or asymptomatic SARS-CoV-2 infection to susceptible animals. 

## Figures and Tables

**Figure 1 animals-12-00614-f001:**
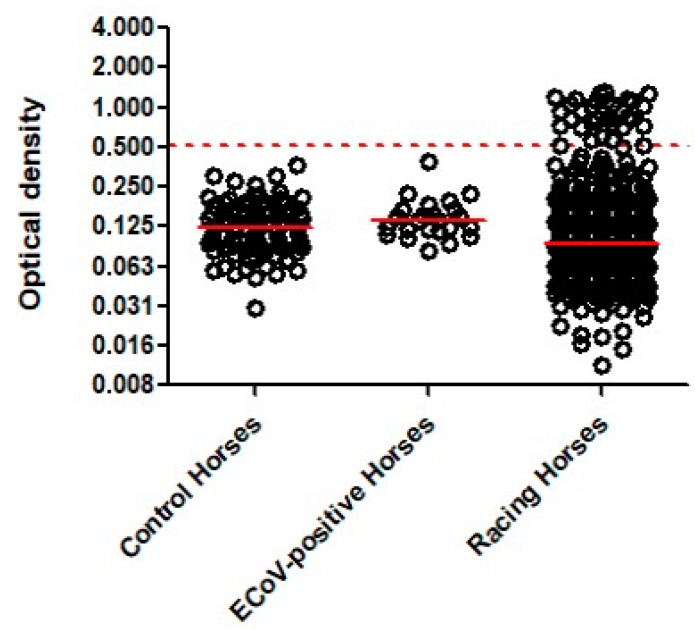
ELISA results from 88 pre-COVID-19 control horses, 24 ECoV-seropositive horses and 633 serum samples collected from 587 racing Thoroughbreds against the recombinant receptor-binding domain of SARS-CoV-2 spike protein. The dashed red line represents the cut-off (0.507). The solid red lines represent the median OD.

**Table 1 animals-12-00614-t001:** Oligonucleotide sequences of primers, probe and positive plasmid control used to detect SARS-CoV-2 by qPCR.

Target Gene (GenBank)	Oligonucleotides
Spike gene (MT773134)	SARS-CoV-2-forward primer: GGCACAGGTGTTCTTACTGAGTCTAAC SARS-CoV-2-reverse primer: CAAGTGTCTGTGGATCACGGAC SARS-CoV-2-probe: FAM-TGGCAGAGACATTGCTGA-MGB Plasmid positive control: TTCAACTTCAATGGTTTAACAGGCACAG GTGTTCTTA CTGAGTCTAACAAAAAGTTTCTGCCTTTCCAACAAT TTGGCAGAGACATTGCTGACACTACTGATGCTGTCCGTGATCCACAGACACTTGAGATTCTTGACATTACACCATGT

## Data Availability

Data available on request due to privacy restrictions.

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
