# Peer review of "Investigation of the Role of Healthy and Sick Equids in the COVID-19 Pandemic through Serological and Molecular Testing"

_animals, 2022, doi:10.3390/ani12050614_

Round 1

Reviewer 1 Report

This manuscript contains important information but I do have some suggestions for improvement.

  1. I recommend including an author who was intimately involved with the SARS-CoV-2 outbreak in the racetrack personnel as there is currently no citation given for that outbreak throughout this manuscript. It would strengthen the paper to include more details such as the variant of SARS-CoV-2 implicated in the outbreak. Were all infected people truly asymptomatic?
  2. In the introduction, the Damas publication is mentioned but it should be more clearly stated that their results show that many mammals fall into each binding score category (ranging from low to high) and that equus caballus and equus asinus both appear to fall into the 'low' binding score category.
  3. The introduction also mentions SARS-CoV-2 documented in dogs, cats, tigers and minks, but the OIE and USDA/APHIS (as of Dec 6, 2021) list many more species that have been documented with SARS-CoV-2 infections to date (pet ferrets, puma, snow leopards, gorillas, otters, white-tailed deer, binturong, fishing cat, coatimundi, hyenas, lynx and a hippopotamus...worth mentioning for completeness.
  4. In the materials and methods section, the racetrack outbreak is mentioned again. Please clarify the statement regarding confidentiality issues...is this in regard to the humans or the horses? The sentence is not clear.
  5. In the discussion, please provide a citation for the following statement: Serology has been shown to be a more accurate diagnostic modality to assess exposure to SARS-CoV-2 compared to antigen-detection via qPCR.
  6. Also in the discussion, please make it very clear that a major limitation of the study involves the inability to test nasal or nasopharyngeal secretions from horses at the racetrack for the virus--which means that virus was not detected and could not be sequenced to confirm that the virus infecting the people at the racetrack was the same virus infecting the horses.

Author Response

1. I recommend including an author who was intimately involved with the SARS-CoV-2 outbreak in the racetrack personnel as there is currently no citation given for that outbreak throughout this manuscript. It would strengthen the paper to include more details such as the variant of SARS-CoV-2 implicated in the outbreak. Were all infected people truly asymptomatic?

During 2020, various outbreaks of COVID occurred among horse racetrack workers in California. The best-described outbreak occurred at Golden Gate Fields among its 563 employee and the outbreak was discovered by the contact tracing efforts of the local health department. The situation at Del Mar Race Track occurred at an earlier time in 2020 and only involved asymptomatic subjects. While local health departments and public healthy units were involved in this outbreak, an institution or research group did not publish the information. While we have contacted members of the San Diego County public health department, nobody has been willing to share information on the variant of SARS-CoV-2. Therefore, the authors are unable to provide the requested information.

2. In the introduction, the Damas publication is mentioned but it should be more clearly stated that their results show that many mammals fall into each binding score category (ranging from low to high) and that equus caballus and equus asinus both appear to fall into the 'low' binding score category.

As suggested by the reviewer, the missing information has been added to the introduction.

3. The introduction also mentions SARS-CoV-2 documented in dogs, cats, tigers and minks, but the OIE and USDA/APHIS (as of Dec 6, 2021) list many more species that have been documented with SARS-CoV-2 infections to date (pet ferrets, puma, snow leopards, gorillas, otters, white-tailed deer, binturong, fishing cat, coatimundi, hyenas, lynx and a hippopotamus...worth mentioning for completeness.

The authors agree with the reviewer that SARS-CoV-2 infections have to date been documented in a variety of animal species. The sentence in questions (line 67) refers specifically to suspected human to animal transmissions of SARS-CoV-2. To the authors’ knowledge, such infections have only been reported in dogs, cats, tigers, lions and minks.

4. In the materials and methods section, the racetrack outbreak is mentioned again. Please clarify the statement regarding confidentiality issues...is this in regard to the humans or the horses? The sentence is not clear.

A link to the reported Del Mar outbreak has been added. The statement of confidentiality relates to the horses as these animals were monitored for the presence of drug residues.

5. In the discussion, please provide a citation for the following statement: Serology has been shown to be a more accurate diagnostic modality to assess exposure to SARS-CoV-2 compared to antigen-detection via qPCR.

The authors would like to clarify that for domestic animals, the use of serology has shown to capture a larger number of infected animals compared to qPCR testing. The authors recognize that laboratory-based qPCR is the recommended test for diagnoses of acute cases and that serological tests are important to define epidemiological questions, such as attack rate. The sentence has been changed to avoid any confusion and a reference has been added.

6. Also in the discussion, please make it very clear that a major limitation of the study involves the inability to test nasal or nasopharyngeal secretions from horses at the racetrack for the virus--which means that virus was not detected and could not be sequenced to confirm that the virus infecting the people at the racetrack was the same virus infecting the horses.

We thank the reviewer for this very important point, which we initially stated as “the inability to test nasal or nasopharyngeal secretions from SARS-CoV-2 by qPCR”. We have further expanded this major limitation as suggested by the reviewer and highlighted that the inability to test the horses and sequence potential SARS-CoV-2 qPCR-positive samples precluded us from determining that horses were infected with the same virus responsible for asymptomatic COVID-19 in humans.

Reviewer 2 Report

Overall, this work by Lawton et al. is well written, flows nicely, and I am pleased to see the release of "negative" data. 

The study is well designed given the absence of information about SARS-CoV-2 in equids. I was impressed with the quantity of samples available for the study and it is clear this was a large amount of work to organize. 

Major comments: 

  • I'm concerned the ELISA may need to be optimized.
    • These samples were run as singlets, and can result in contamination. I understand the realities of trying to run the volume of samples used - is it possible to include a sentence listing this as a limitation.
    • The OD from your negative samples is quite high, and would be considered positive in most instances. The manuscript you cite for the ELISA had an upper limit of ~0.4 for control samples, do you have a hypothesis for why your control samples were so high? Is it possible to either re-run these samples with different control sera or add an explanation in the manuscript? 

Minor comments: 

  • Given the recent studies in white tailed deer, I wonder if it would be useful discussing some of that reverse spillover. They seem to be truly asymptomatic, and may be a nice comparison.
  • There is one manuscript (Bosco-Lauth) that was very recently published describing experimental infection in one horse. However, given that they did not show any data from the one horse and it is a single animal, I will defer to your judgement to if you want to cite this. 
  • Non-critical - you may consider discussing the fact that you only sampled clinical horses, and your data suggests they may not show any clinical signs. Additionally, as you cited a study talking about pet-human interaction contributing to transmission, it may be interesting to discuss that racehorses are typically handled differently than privately owned "pet" horses, and speculate on if this could change transmission patterns. 

Author Response

  1. I’m concerned that the ELISA may need to be optimized.

The ELISA validated for this study uses commercial recombinant SARS-CoV-2 RBD of the spike protein and was addapted from a previous published assay used to screen against SARS-CoV-2 infection in cats and dogs. S protein, anti-horse IgG horseradish peroxidase conjugate and serum dilutions were determined prior to assay validation using standard checkboard titration procedures. The authors acknowledge that without longitunial serum samples from documented horse infections with SARS-CoV-2, it may be difficult to determine true accuracy of the assay. Due to the lack of positive goldstandard samples, the assay was validated using pre-pandemic serum samples from healthy horses as well as convalescent serum samples from horses with documented ECoV infection. Due to moderate background noise of the negative equine samples, the OD values of these samples had a wide range 0.973 to 2.391 (median 1.542), which translated in a high cutoff value for a positive result. With the established cutoff, the authors determined that 41 serum samples from 36 horses tested seropositive against SARS-CoV-2. While the results showed the presenced of antibodies to the SARS-CoV-2 S protein in equids, it is possible due to to the high OD, that in the study population a larger number of horses may have displayed levels of specific antibodies below the cutoff value. One interesting aspect is that aamongst 32 horses with a single SARS-CoV-2 seropositive sample, 4 horses showed seroconversion between two sample collection time points, emphasizing the rise in SARS-CoV-2 specific antibody levels. The authors also acknowledge the limitation that the positive ELISA results were not confirmed via serum neutralization. Serological screening of SARS-CoV-2 infection in cats and dogs using various serological platforms has established that animals testing positive for RBD ELISA and VN assay are confirmed positive, while animals testing positive for RBD ELISA and negative by VN assay are defined as suspect. Various publications have shown individual differences in the development of neutralizing antibodies relating to different levels of SARS-CoV-2 exposure and time of sampling following infection. In recent studies in humans and cats with asymptomatic or mild infection with SARS-CoV-2, samples were seropositive by ELISA but failed to neutralize virus infection (Zhao et al. 2021; GeurtsvanKessel et al., 2020). The present manuscript has been edited in order to reflect that the RBD ELISA seropositive horses are considered “suspect positive” and the limitation of not using  a confirmatory serological assay was added in the study limitations.

  1. These samples were run as singlets, and can result in contamination. I understand the realities of trying to run the volume of samples used - is it possible to include a sentence listing this as a limitation.

We thank the reviewer for mentioning the limitation of running the samples in singlets. While not reported in the study, positive (41 samples) and negative control samples (88 samples) with sufficient plasma available were re-run to confirm the results, which showed 100% agreement. As suggested by the reviewer the limitation of using singlets was added in the discussion.

  1. The OD from your negative samples is quite high, and would be considered positive in most instances. The manuscript you cite for the ELISA had an upper limit of ~0.4 for control samples, do you have a hypothesis for why your control samples were so high? Is it possible to either re-run these samples with different control sera or add an explanation in the manuscript? 

The authors believe that the background noise observed in the 88 pre-COVID-19 serum samples may related to the storage time (6 years) despite the fact that the samples were properly stored at -80°C. Additional information was included in the discussion regarding the high OD value for a positive serum sample.

  1. Given the recent studies in white tailed deer, I wonder if it would be useful discussing some of that reverse spillover. They seem to be truly asymptomatic, and may be a nice comparison.

As mentioned by the reviewer, the high detection rate of SARS-CoV-2 in free ranging white-tailed deer is very concerning and suggest that deer may act as a potential reservoir host with potential reverse spillover to humans. While the susceptibility to infection in white-tailed deer is high and animal-to-animal transmission has been documented, it does not appear that SARS-CoV-2 infection in equids elicits viral shedding enough to infect susceptible humans. However, experimental challenge studies are needed in order to study the clinical, hematological, molecular, and serological features of adult horses infected with SARS-CoV-2.

  1. There is one manuscript (Bosco-Lauth) that was very recently published describing experimental infection in one horse. However, given that they did not show any data from the one horse and it is a single animal, I will defer to your judgement to if you want to cite this. 

Many thanks for mentioning the recent letter from Angela Bosco-Lauth and collaborators. We have added that reference in the discussion.

  1. Non-critical - you may consider discussing the fact that you only sampled clinical horses, and your data suggests they may not show any clinical signs. Additionally, as you cited a study talking about pet-human interaction contributing to transmission, it may be interesting to discuss that racehorses are typically handled differently than privately owned "pet" horses, and speculate on if this could change transmission patterns. 

Indeed, the authors mentioned the fact that the first part of the study focused on horses with acute onset of fever and respiratory signs with not a single horses testing qPCR-positive for SARS-CoV-2. The authors proposed that the reason for negative SARS-CoV-2 qPCR results in the present study population might relate to disease expression, i.e. horses apparently do not express clinical disease. While the evidence for silent infection in equids is limited, various studies using Italian trotters and one experimental horse infection, showed no susceptibility of horses to SARS-CoV-2. It would be of interest to sample healthy horses with contact to symptomatic and asymptomatic COVID-19 horse owners to determine the impact of spillover infection. It would be speculative to determine if racehorses or privately owned horses are at greater risk of becoming infected with SARS-CoV-2 based solely on husbandry and exercise practices. One would have to compare prevalence factors such as stabling type, stall cleaning practices, feeding practices and training/ridding time and practices in order to determine which population is at greater risk of becoming infected with SARS-CoV-2. The authors speculate that the management and husbandry practices used around horses (mostly relating to human-to-horse contact time) does in general not promote SARS-CoV-2 transmission between SARS-CoV-2 infected humans and equids.

Reviewer 3 Report

The manuscript presents the results of a cross-sectional investigation for SARS-CoV-2 in horses in the USA. Respiratory samples from animals with respiratory disease, previously analysed for equine parvovirus, showed no presence of SARS-CoV-2 RNA by real-time RT-PCR. In contrast, a number (6.1%) of equine sera collected in late 2020 were positive by an ELISA test based on the receptor binding domain (RBD) of SARS-CoV-2 spike protein.

Critique

The main drawback of the paper is that sera testing positive by the ELISA test were not confirmed by the gold standard, which is serum neutralisation (SN). Analogous to what reported in humans seropositive for HCoV-OC43, antibodies against the related equine coronavirus (ECoV) may cross-react with SARS-CoV by ELISA. Accordingly, pre-pandemic sera and sera from ECoV infected horses displayed exceptionally high OD values by the SARS-CoV-2 ELISA, which forced the authors to set up a very high cutoff value (≥3.3). Therefore, without a confirmatory assay the results of the present paper cannot be considered reliable. As an additional concern, t is not clear why the authors used an in-house real-time RT-PCR assay to detect SARS-CoV-2 RNA, while a plethora of assays have been already standardised and published. There is no standardisation of their assay in terms of analytical sensitivity, specificity and reproducibility.

Author Response

  1. The main drawback of the paper is that sera testing positive by the ELISA test were not confirmed by the gold standard, which is serum neutralisation (SN). Analogous to what reported in humans seropositive for HCoV-OC43, antibodies against the related equine coronavirus (ECoV) may cross-react with SARS-CoV by ELISA. Accordingly, pre-pandemic sera and sera from ECoV infected horses displayed exceptionally high OD values by the SARS-CoV-2 ELISA, which forced the authors to set up a very high cutoff value (≥3.3). Therefore, without a confirmatory assay the results of the present paper cannot be considered reliable.

The authors agree with the reviewer that a reference standard serological assay would be ideal to support the presence of SARS-Cov-2-specific antibodies. Unfortunately, serum neutralization testing has remained unavailable in most veterinary diagnostic research laboratories. Serological screening of SARS-CoV-2 infection in cats and dogs using various serological platforms has shown that animals testing positive for RBD ELISA and VN assay are confirmed positive, while animals testing positive for RBD ELISA and negative by VN assay are defined as suspect. Various publications have shown individual differences in the development of neutralizing antibodies relating to different levels of SARS-CoV-2 exposure and time of sampling following infection. In recent studies in humans and cats with asymptomatic or mild infection with SARS-CoV-2, samples were seropositive by ELISA but failed to neutralize virus infection (Zhao et al. 2021; GeurtsvanKessel et al., 2020). The present manuscript has been edited in order to reflect that the RBD ELISA seropositive horse are considered “suspect positive” and the limitation of not using  a confirmatory serological assay was added in the study limitations.

  1. As an additional concern, it is not clear why the authors used an in-house real-time RT-PCR assay to detect SARS-CoV-2 RNA, while a plethora of assays have been already standardised and published. There is no standardisation of their assay in terms of analytical sensitivity, specificity and reproducibility.

The authors’ laboratory has ample experience with the design and validation of qPCR assays for the detection of animal pathogens. The SARS-CoV-2 assay was in the authors’ opinion properly validated and the information was reported in the manuscript. The primer/probe are continuously been blasted with new genetic information pertaining to all SARS-CoV-2 variants in order to comply with assay specificity. While the authors agree that the assay was not validated using true biological samples, standard curve, amplification efficiency and detection limit were generated using plasmid containing the target sequence as shown in Table 1.  

Round 2

Reviewer 3 Report

The revised manuscript has not sufficiently addressed this reviewer's concern about the reliability of the obtained data. Without any confirmatory assay (i.e., serum neutralization), the obtained data and their interpretation have no solid basis.